# Current Knowledge on Radiation-Therapy-Induced Erectile Dysfunction in Prostate-Cancer Patients: A Narrative Review

Connie Labate [1,2,†], Andrea Panunzio [1,†], Francesco De Carlo [1,*], Federico Zacheo [1], Sara De Matteis [3], Maria Cristina Barba [3], Umberto Carbonara [2,4], Floriana Luigina Rizzo [1], Silvana Leo [5], Saverio Forte [2], Pasquale Ditonno [2], Alessandro Tafuri [1] and Vincenzo Pagliarulo [1]

1   Department of Urology, "Vito Fazzi" Hospital, 73100 Lecce, Italy; connielabate2903@gmail.com (C.L.); panunzioandrea@virgilio.it (A.P.); federico.zacheo@gmail.com (F.Z.); florianarizzo@libero.it (F.L.R.); aletaf@hotmail.it (A.T.); enzopagliarulo@yahoo.com (V.P.)
2   Department of Emergency and Organ Transplantation, Urology and Andrology Section, University of Bari, 70124 Bari, Italy; u.carbonara@gmail.com (U.C.); pasquale.ditonno@uniba.it (P.D.); saverio.forte@gmail.com (S.F.)
3   Department of Radiation Therapy, "Vito Fazzi" Hospital, 73100 Lecce, Italy; sarademat.89@gmail.com (S.D.M.); crissile80@gmail.com (M.C.B.)
4   Department of Urology, Royal Surrey NHS Trust, London NW3 2PS, UK
5   Department of Oncology, "Vito Fazzi" Hospital, 73100 Lecce, Italy; silvileo59@gmail.com
*   Correspondence: francescodecarlo03@yahoo.it; Tel.:+39-0832661324; Fax: +39-0832661382
†   These authors contributed equally to this work.

**Abstract:** Prostate cancer is the most frequently diagnosed cancer in men in the United States. Among the different available treatment options, radiation therapy is recommended for localized or even advanced disease. Erectile dysfunction (ED) often occurs after radiation therapy due to neurological, vascular, and endocrine mechanisms resulting in arterial tone alteration, pudendal-nerve neuropraxia, and lastly fibrosis. Considering the influence of quality of life on patients' treatment choice, radiation-therapy-induced ED prevention and treatment are major issues. In this narrative review, we briefly summarize and discuss the current state of the art on radiation-therapy-induced ED in PCa patients in terms of pathophysiology and available treatment options.

**Keywords:** erectile dysfunction; radiation therapy; prostate cancer; sexual outcomes; sexual rehabilitation

## 1. Introduction

Prostate cancer (PCa) is the most frequently diagnosed cancer and the second most common cancer-related cause of death in men in the United States [1]. The standard of care for clinically localized PCa is still debated in terms of efficacy and side effects [2]. In this context, radiation therapy (RT) represents an established definitive treatment option for localized or even locally advanced disease [2]. However, radiation-therapy-induced erectile dysfunction (RI-ED) is often reported by patients leading to significantly decreased sexual quality of life (QoL) [3]. Newly diagnosed ED occurs approximately in 30–40% of patients undergoing RT in the early period; these rates further increase during the following two years [4]. Therefore, considering the influence of QoL on patients' treatment choice in the context of an increasing population of PCa survivors and the long-term treatment-related side effects, RI-ED prevention and treatment are major issues. The aim of this narrative review is to briefly summarize and discuss the current state of the art on RI-ED in PCa patients in terms of pathophysiology and available treatment options.

## 2. Radiation-Therapy-Induced Erectile Dysfunction Pathophysiology

Several neurological, vascular, and endocrine mechanisms related to RI-ED have been proposed, with arterial damage and neurovascular exposure to high-dose radiation as the main investigated causes, and the internal pudendal artery (IPA), the prostatic neurovascular plexus (PNP), and ejaculatory ducts as the predominant structures at risk [5–7].

Pudendal and cavernous nerves are just minimally and indirectly affected by RT [5]. Radiation therapy induces a proinflammatory cytokine cascade that promotes the development of an inflammatory microenvironment and thereby neurovascular toxicity; the inflammation extent is directly proportional to the amount of irradiated prostatic tissue, fraction delivery time, patient setup errors, and the use of rectal sparing protocols [5,8,9]. Furthermore, there is also evidence of endothelial damage and accelerated atherosclerosis of the IPA, deep dorsal, main cavernous, and bulbar veins, and the prostatic venous plexus resulting in arterial occlusive disease of the IPA and abnormal blood flow, as reported in 40–85% of patients [5,10–12]. Canine model studies also showed morphological damage (lymphocytic cuffing and hyaline change of prostate-gland vessels, reactive perivascular fibrosis, widespread interstitial fibrosis, and the atrophy of peripheral nerves with moderate axon loss) responsible for arterial tone alteration and pudendal-nerve neuropraxia, which causes reduced oxygenation and structural changes in the corpora cavernosa, lastly resulting in fibrosis [12].

The radiation-induced damage of genital nerves certainly contributes to ED both directly and indirectly, affecting the early stage of reflexogenic penile erection and the fibrosis that is induced within three years post-RT in the pelvic-floor muscles (PFMs) [13–17]. Unfortunately, the existing literature lacks data analyzing the potential PFM damage due to prostate irradiation. Ribeiro et al. performed the morphofunctional assessment of PFM after RT via magnetic resonance imaging (MRI), surface electromyography (EMG), and digital rectal palpation, revealing a decrease in EMG activity, weak voluntary contractions, and a higher prevalence of pelvic complaints in men who had undergone RT; conversely, no significant changes emerged among the morphofunctional parameters evaluated with MRI [18]. Penile erection also requires a psychogenic route triggered by sexual thoughts and stimuli. It is quite clear that psychiatric mechanisms may also contribute to ED occurring in PCa patients due to depression and anxiety related to their diagnosis, and frustration, shame, and lack of confidence about sexual performance related to subsequent treatments [19–22]. Lastly, endocrinological alterations resulting from androgen deprivation therapy (ADT) administration may also contribute to RI-ED. Several studies reported significantly higher rates of ED after RT in men who had also received ADT compared to those of patients who had not, primarily due to iatrogenic testosterone deficiency, which led to decreased sexual desire, decreased erectile function, anejaculation, and delayed orgasm [23–26].

### 3. Radiation Therapy Options in PCa Patients and Associated RI-ED Rates

#### 3.1. External Beam Radiation Therapy

An overview of the main studies evaluating the impact of different types of RT on erectile function is reported in Table 1. Intensity-modulated RT (IMRT) and volumetric modulated arc therapy (VMAT) with image-guided RT (IGRT) are currently accepted as the best available approaches for external beam RT (EBRT) [2]. Both techniques are indicated in patients presenting from low- to high-risk disease and employ dynamic multileaf collimators that automatically and continuously adapt to the target volume [2]. The probability of maintaining useful erectile function for sexual intercourse at two years after EBRT depends on the patient's baseline erectile function, assessed through validated questionnaires (e.g., International Index for Erectile dysfunction, IIEF-5; Prostate Cancer Symptom Indices, PCSIs). Depending on the reports, patients who present excellent baseline erectile function have a 70% probability of preserving it after treatment [27]. Dose escalation using EBRT provides treatment intensification that shows consistent improvement in biochemical failure [28]. The ProtecT trial compared active surveillance, radical prostatectomy, and dose-escalated, conventionally fractionated IMRT in localized PCa; the study proved a 40% erectile function decline with dose-escalated RT, superior erectile function preservation after IMRT over radical prostatectomy, and a later ED onset after RT compared to those of radical prostatectomy [29]. The NRG/RTOG0126 trial randomized patients to either 70.2 Gray (Gy) in 39 fractions or 79.2 Gy in 44 fractions to assess the impact of

dose-escalated EBRT delivered using IMRT on ED occurrence, reporting no statistically significant difference between treatment arms [28]. Additionally, new radiologic tissue-centering methods are contributing to RI-ED rate decreases. The involvement of magnetic resonance imaging in the development of modern target EBRT techniques was effective in preserving erectile function due to the contouring of neurovascular bundles (NVBs) and IPAs on pretreatment [30].

### 3.2. Hypofractionated Radiation Therapy

Considering the high sensibility of tumor cells to an increased dose per fraction, hypofractionation (HFX) develops fewer fractions with larger doses per fraction to induce DNA damage and PCa cell death, requiring tighter prostatic margins to reduce toxicity in normal regional tissue [2]. Hypofractionation with 2.5–3.4 Gy/fraction is moderate HFX, while HFX with >3.4 Gy/fraction is ultra-HFX [2]. Several trials proved the safety and efficacy of HFX in disease control [31–34]. The HYPO-RT-PC trial compared the outcomes of patients receiving conventional fractionation RT (78.0 Gy in 39 fractions, 5 days per week for 8 weeks) and ultra-HFX RT (42.7 Gy in 7 fractions, 3 days per week for 2.5 weeks), demonstrating no clinically or statistically significant differences between the two groups in terms of genitourinary symptoms or sexual deterioration. Although there were higher acute toxicity rates among ultra-HFX RT-treated patients, mainly due to gastrointestinal symptoms, ultra-HFX was as well-tolerated as conventional fractionation up to 6 years after the completion of treatment [35]. Rasmusson et al. evaluated the risk of ED with a particular focus on the impact of radiation doses on the penile base, demonstrating that age at RT was the strongest predictor of ED, followed by a near-maximal dose to the penile base [36]. A recent meta-analysis compared conventional or moderate and ultra-HFX protocols, showing no differences between these RT options in terms of both acute and late genitourinary toxicity [37]. Additionally, the use of intraprostatic fiducials was particularly useful, leading to toxicity reduction and more selective prostate RT [38–42]. For example, Pepe et al. demonstrated the effectiveness of injecting a hydrogel spacer (SpaceOAR) in preserving permanent sexual function in 65.2% of cases at a median follow-up of 6 and 18 months [43]. Lastly, stereotactic body RT (SBRT) is an extreme form of HFX where treatment is usually administered in one to five fractions and is considered a valid alternative to conventional or moderate HFX in low- and intermediate-risk PCa, although its use was also evaluated for high-risk patients [2]. In this context, two randomized Phase 3 clinical trials reported no significant differences in safety between conventionally fractionated or moderate HFX and SBRT [44,45].

### 3.3. Proton-Beam Therapy

Lastly, proton-beam therapy (PBT) exploits the sharper dose fall off of protons beyond their deposition depth and tendency to deposit almost their entire radiation dose at the end of the particle's path to prevent normal peripheral tissue from being damaged. Data on sexual toxicity related to PBT are still limited, although erectile function reduction was observed from 90% at baseline to 62% and 67% at 1 and 5 year(s) of follow-up, respectively [46].

**Table 1.** Overview of studies evaluating radiation therapy for prostate cancer and its impact on erectile function.

| Name/Author | Year | Country | Study Design | F/U | Participants Number | Therapeutic Option | Results |
|---|---|---|---|---|---|---|---|
| Hall et al. [28] NRG/RTOG 0125 trial | 2022 | USA | Prospective Phase III RCT | 24 months | 1532 | EBRT | ED occurrence |
| | | | | | 769 | 70.2 Gy in 39 fractions | 38.1% |
| | | | | | 763 | 79.2 Gy in 44 fractions | 49.7% |
| | | | | | | | *p* = 0.051 |
| Donovan et al. [29] ProtecT trial | 2016 | UK | Comparative trial | 6 years | 545 | AS | ED occurrence/worsening 15% |
| | | | | | 553 | RP | 55% |
| | | | | | 545 | EBRT | 45% |
| Dearnaley et al. [31] CHHiP trial | 2016 | UK, Ireland, Switzerland, New Zealand | International multicenter phase III RCT | 5 years | 3216 | EBRT | Sexual symptoms ≥2 LENT-SOMA scale |
| | | | | | 1065 | CF 74 Gy | 67% |
| | | | | | 1074 | HFX 60 Gy | 65% |
| | | | | | 1077 | HFX 57 Gy | 64% |
| Lee et al. [33] NRG Oncology RTOG0415 | 2016 | USA | Randomized Phase III noninferiority comparing study | 5.8 years | 1092 | EBRT | GU toxicity, early and late |
| | | | | | 542 | CF 73.8 Gy | 61.6%, 52.3% |
| | | | | | 550 | HFX 60 Gy | 62.2%, 58.1% |
| Catton et al. [34] PROFIT trial | 2017 | Canada, Australia, France | Noninferiority RCT | 6 years | 1206 | EBRT | Late ≥ 3 GU toxicity |
| | | | | | 608 | HFX 60 Gy | 2.1% |
| | | | | | 598 | CF 78 Gy | 3.0% |
| Rasmusson et al. [36] HYPO-RT-PC trial | 2020 | Sweden | Open-lab Phase III RCT | 5 years | 673 | EBRT | ED occurrence |
| | | | | | 330 | CF | 27% |
| | | | | | 343 | UHF | 27% |
| Pepe et al. [42] | 2022 | Italy | Experimental trial | 18 months | 56 | Hydrogel injection SpaceOAR before HFX 60 Gy | EF preservation 62.5% |
| Brand et al. [45] PACE-B trial | 2022 | UK, Ireland, Canada | Open-label, multicohort, randomized, controlled, Phase III trial | 2 years | | | RTOG grade ≥2 GU toxicity |
| | | | | | 430 | CF 78 Gy | 2% |
| | | | | | 414 | SBRT 36 Gy | 13% |
| Ho et al. [46] | 2018 | USA | Observational trial | 5 years | 254 | PBT | Ability to function sexuality loss (EPIC) 1 year F/U 24% 3 year F/U 82% 5 year F/U 54% |

Abbreviations: RCT, randomized clinical trial; EBRT, Gy, Gray; external beam radiation therapy; ED, erectile dysfunction; AS, active surveillance; RP, radical prostatectomy; CF, conventional fractionated; HFX, hypo fractionated; LENT-SOMA, Late Effect Normal Tissue Task Force—Subjective Objective, Management an Analytic; GU, genitourinary; UHF, ultra-hypo-fractionated; SBRT, stereotactic body radiation therapy; PBT, proton-beam therapy; EPIC, expanded prostate cancer index composite.

## 4. Sexual Rehabilitation in RI-ED Patients

The international guidelines' specific recommendations for penile RI-ED rehabilitation in PCa patients are lacking [47]. All types of ED share the same pathophysiological mechanisms, but given the different leading causes (direct radiation effect on nerves and vessels), unequal damage severity and treatment response are also conceivable, and a specific therapeutic algorithm could be proposed. Moreover, the use of proerectile drugs may prove useful not just in RI-ED treatment, but also in its prevention, avoiding cavernous fibrosis and maintaining penile length. An overview of the main studies evaluating therapeutic options and outcomes for RI-ED is reported in Table 2.

### 4.1. Phosphodiesterase Type 5 Inhibitors

Phosphodiesterase Type 5 inhibitors (PDE-5i) represent the first line of treatment in ED. Their mechanism of action consists in relaxing smooth muscle, consequently increasing blood flow and compressing the subtunical venous plexus, resulting in erection [2,48].

Several studies demonstrated the efficacy of PDE-5i in improving erectile function in approximately 70 to 91% of patients with different degrees of ED after RT [49–55]. For example, according to Zelefky et al., daily therapy of 50 mg sildenafil could delay acute sexual toxicity, preventing the occurrence of moderate/severe ED within 12 months after RT in 73% of patients, whereas no differences were found at 24 months compared with the control group [52]. Similarly, Ilic et al. reported no long-term differences between sildenafil and placebo regarding erectile function, although men in the sildenafil group exhibited significantly better IIEF-5 scores at 4 weeks and 6 months [56]. Lastly, the randomized Therapy Oncology Group [0831] clinical trial showed the effect of a daily therapy of 5 mg of tadalafil on a cohort of 121 patients who had undergone EBRT or RP, reporting no significant differences in the study group compared with the placebo at 30 weeks and 12 months of follow-up [57]. The on-demand use of tadalafil was also investigated, reporting no differences in terms of erectile-function improvement compared with a daily dose of 5 mg [58–60].

### 4.2. Vasoactive Injectables, Vacuum Therapy, and Pelvic-Floor Physiotherapy

The efficacy and safety of other treatment options for ED rehabilitation such as vasoactive-agent intracavernosal injections and regenerative therapies in the specific context of PCa patients treated with RT have never been tested in prospective controlled or even retrospective studies. However, different contributions have included these patients, demonstrating interesting results. Vasoactive-agent intracavernosal-injection therapy represents a treatment option in case of PDE-5i failure and is more likely to be effective, especially in cases of neuropraxia [61]. Available intracavernosal-injection therapies include alprostadil, papaverine, and phentolamine [2,61]. Alprostadil can be administered as a cream (200 and 300 µg), via the urethral meatus, or via intraurethral insertion as a Medicated Urethral System for Erection medicated pellet (MUSE™) (125–1000 µg) [47].

Vacuum erection devices (VEDs) provide mechanic erections obtained via corporal and gland venous-blood infarction [62]. A recent meta-analysis revealed the effectiveness of VEDs in preserving vascular and ejaculatory ductal potency during and after RT [63]. Additionally, VEDs could improve the effect of PDE5i during an acute neurotoxicity state [12,63]. The training of PFMs and muscles indirectly related to the pelvis such as abdominal and gluteal muscles in the context of a multidisciplinary approach, also including back manometric biofeedback, electrotherapy, and vacuum pumps combined with biofeedback, may represent a promising alternative to pharmacological treatments, although there is a need for more rigorously designed randomized clinical trials [64].

Lastly, semirigid and inflatable penile prostheses constitute the last treatment option in the case of the failure or refusal of previous therapy [65].

### 4.3. Regenerative Therapies and Future Directions

Recently, emerging treatments have been investigated in both clinical and preclinical settings, such as low-intensity shock wave therapy (Li-SWT), platelet-rich plasma (PRP) penile injections, and stem-cell therapy [66]. These options are regenerative therapies because they could potentially reverse or halt the ED process [67]. Specifically, Li-ESWT is recommended by guidelines for either mild or moderate ED due to its mechanism of promoting cellular apoptotic alteration and tissue neovascularization through the release of various angiogenic factors [47,68,69]. In the preclinical studies of animal models, LI-ESWT was effective in inducing neuroprotection and nerve regeneration, and in improving cavernosal blood flow; no clinical studies have focused on LI-ESWT for RI-ED, whereas improvements in IIEF-5 score were reported after radical prostatectomy [70,71].

Platelet-rich plasma (PRP) is an autologous blood-derived product consisting of concentrated platelets and cellular growth factors, and obtained through patient blood centrifugation and platelet activator addiction [72,73]. PRP administration is effective in modulating the inflammatory response, inducing neuronal and vascular regeneration, and

reducing tissue fibrosis [74]. Both animal and human studies produced hopeful results in terms of sexual outcomes [75–78].

Radiation mitigators such as SS-TGF receptor inhibitors (e.g., small molecules, antibodies), gene therapy (activators of the nitrergic–neural system, endothelial growth factor promoters, and modulators of ion channels in smooth muscle cells) and stem-cell therapy represent new and promising research fields [79–85]. Assuming that oxidative damage involved in local proinflammatory cytokine cascades plays a key role in the development of radiation-induced fibrosis, the administration of radiation mitigators may be useful in preventing or inhibiting radiation-induced fibrosis due to their ability to mitigate the inflammatory cascade, as tested in preclinical animal models. In this context, the transforming growth factor also plays a crucial role in tumor survival, although the optimal timing to achieve tumor inhibition through the administration of SS-TGF receptor inhibitors without compromising radiation cancer control is still not clear [79–81].

On the basis of current evidence [47], a proposal for rehabilitative treatment lines in RI-ED patients is illustrated in Figure 1.

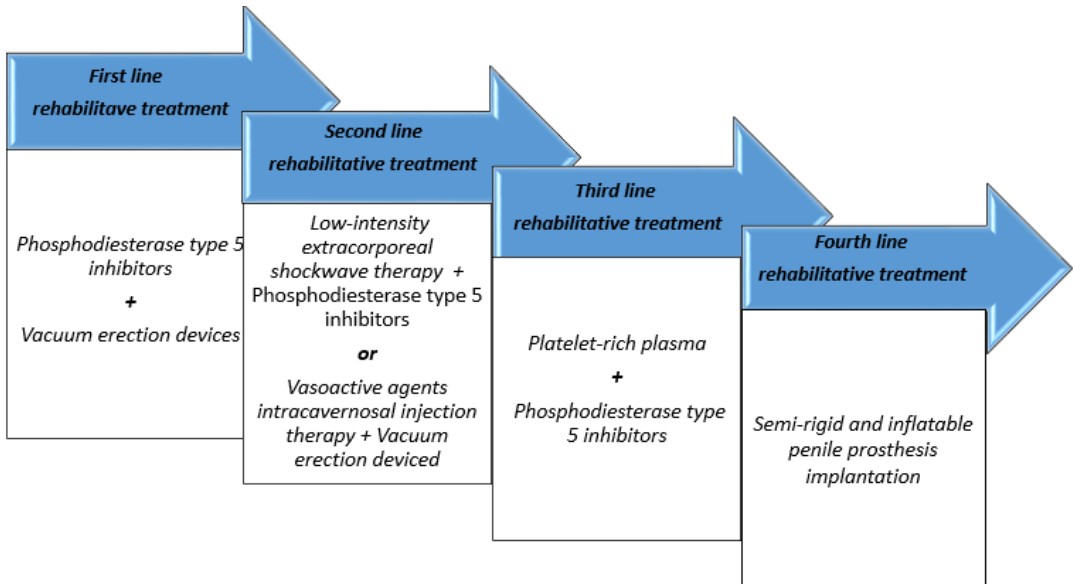

**Figure 1.** Suggested rehabilitative treatment lines in patients with radiation-induced erectile dysfunction [47].

**Table 2.** Overview of the studies evaluating the therapeutic options and outcomes for radiation-induced erectile dysfunction.

| Name/Author | Year | Country | Study Design | Participants Number | Therapeutic Option | Dosing Schedule | Interval from RT | Outcomes |
|---|---|---|---|---|---|---|---|---|
| S. Kedia et al. [34] | 1999 | USA | Prospective observational trial | 21 | Sildenafil | 50 mg with a titration to 100 mg if needed; 1 sildenafil tablet approximately 1 h before sexual activity | 24.6 +/− 5.8 months | Mean duration of vaginal intercourse (CCEF), 12.7 ± 2.5 min; mean frequency of penetration score (IIEF 5), from 1.3 to 4.0; maintenance of erection score (IIEF 5), from 1.1 to 3.9. |
| Weber DC. et al. [35] | 1999 | Switzerland | Prospective observational trial | 35 | Sildenafil | 100 mg orally once a week for 6 consecutive weeks | 18.5 months | Response rate, from 40% during the first week to 77% at 6 weeks; mean weekly IIEF 5 score of 13.8, 16.0, 17.0, 16.8, 17.0, and 17.6 at weeks 1 to 6, respectively. |
| Zelefky MJ et al. [36] | 1999 | USA | Prospective observational trial | 50 | Sildenafil | Patients were initially given 50 mg of sildenafil and instructed to use the medication on at least three occasions | 19 months | Erection firmness improvement: significant in 74%, partial in 4%, no response in 22%; erection durability improvement: significant in 66%, partial in 6%, no improvement in 28%; libido improvement: in 18%. |
| Watkins Bruner et al. [39] | 2013 | USA | Randomized, double-blinded, placebo-controlled crossover trial | 115 | Sildenafil | 12 weeks of sildenafil or placebo, followed by 1 week of no treatment, 12 weeks of a alternative flexible dosing schedule starting with a 50 mg dose (1 pill) 1 h prior to desired sexual activity and increasing up to 100 mg (2 pills) daily as needed. | 12 months (range 5.5–48) | 66%, any response; 10%, both placebo and sildenafil response; 21%, only sildenafil response; 3%, only placebo response. |
| Ilic D. et al. [40] | 2012 | Australia | Randomized, double-blinded, placebo-controlled trial | 27 | Sildenafil | Standard 50 mg sildenafil tablets, one tablet each night. Patients were reviewed after 1 month, and if no adverse effects had been noted, they were instructed to take 2 tablets each evening. Six-month trial period. | 1 month | Based on IIEF-5 measure, sildenafil vs. placebo: Baseline: 24 both; 4 weeks: 24–21; 12 weeks: 24–20; 2 years: 19–20. |

**Table 2.** *Cont.*

| Name/Author | Year | Country | Study Design | Participants Number | Therapeutic Option | Dosing Schedule | Interval from RT | Outcomes |
|---|---|---|---|---|---|---|---|---|
| Pisansky et al. [41] Therapy Oncology Group [0831] | 2014 | USA Canada | Stratified, placebo-controlled, double-blind, parallel-group study with 1:1 randomization. | 242 | Tadalafil | 5 mg for 24 consecutive weeks | 1 week after RT initiation | EEF5-based EF. Tadalafil vs. placebo. Baseline, 24.8–25.1; 30 weeks, 20.7–20.9; 50 weeks, 21.2–20.4. |
| Incrocci et al. [42] | 2006 | Netherland | Randomized, double-blind, placebo controlled, cross-over-study | 358 | Tadalafil | 20 mg on demand tadalafil or placebo for 6 weeks. | 12 months | Erectile function, IIEF score: Baseline, 8.4; after tadalafil, 17.7; after placebo, 9.5. Erectile function, SEP diary, tadalafil vs. placebo: Question 1: 64–30; Question 2: 47–19; Question 3: 46–12; Question 4: 43–7; Question 5: 48–15. |
| Ricardi et al. [43] | 2010 | Italy | Randomized comparative study | 52 | Tadalafil | On-demand 20 mg or once-a-day 5 mg tadalafil for 12 weeks | 6 months | IIEF-based EF. On-demand 20 mg vs. once-a-day 5 mg: Baseline: 6–6; 1 month: 22–24; 3 months: 25–27. |

Abbreviation: RT, radiation therapy; CCEF, Cleveland Clinic Erectile Function; IIEF-5, International Index of Erectile Function-5; EF, erectile function; SEP, Sexual Encounter Profile.

## 5. Conclusions

Following the substantial progress in its early detection and treatment, PCa has become a highly curable disease, with many patients diagnosed at a clinically localized stage and at a younger age. Although an active treatment such as RT can be effective on cancer control outcomes, ED is a common side effect reported by these patients that significantly affects their sexual QoL. Several available options for penile erectile-function rehabilitation exist that safe and effective in appropriately selected patients. However, further advances are still required in RI-ED prevention and treatment.

**Author Contributions:** C.L., F.D.C., A.P., A.T. and V.P.: project development; C.L. and A.P.: manuscript writing and editing; F.D.C., F.Z., S.D.M., M.C.B., U.C., F.L.R., P.D., S.L., S.F., A.T. and V.P.: critical revision for important intellectual content. All authors have read and agreed to the published version of the manuscript.

**Funding:** This research received no external funding.

**Institutional Review Board Statement:** Not applicable.

**Informed Consent Statement:** Not applicable.

**Data Availability Statement:** Not applicable.

**Conflicts of Interest:** The authors declare no conflict of interest.

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
