# Peer review of "Current Knowledge on Radiation-Therapy-Induced Erectile Dysfunction in Prostate-Cancer Patients: A Narrative Review"

_2673-4397, doi:10.3390/uro3020013_

Round 1
Reviewer 1 Report
The referee would first note that the uro template is not used in this manuscript.
On abstract:
Do not use the form "is the second most frequently diagnosed cancer" but something like "was the second most frequent in 2020" since the difference to lung cancer is 0.2 percentage point and PC may be the most frequent cancer any year now.
"of the art on radiation therapy..." -> of the state
Introduction:
Same "is" to "as" issue than in abstract.
ED does not necessarily lead to decreased quality of life. It certainly has shown to decrease sexual quality of life but total quality of life in comparison to healthy peers seems to be preserved (see "Health-related Quality of Life of Patients Treated With Different Fractionation Schedules for Early Prostate Cancer Compared to the Age-standardized General Male Population" by Reinikainen et al. for example)
Also many of the patients are elderly, and the prevalence of ED prior treatment is also high. Prevalences of prior treatment should be given if such are found. Also it should be clarified how 40 % is measured (does it include all patients or only those that did not have ED prior treatment).
"current state of the art on" --> "current state of".
Radiation therapy induced erectile dysfunction pathophysiology:
What should be clarified is that pudendal nerve does not carry parasympathetic fibers which are the main physiological mechanism for erection. Cavernous nerves stem from sacral roots and are not directly in RT treatment field except for small branches. These should be discussed (see "Prostatic irradiation-induced sexual dysfunction: a review and multidisciplinary guide to management in the radical radiotherapy era (Part I defining the organ at risk for sexual toxicities)" by Ramirez-Fort et al. for example. Of course pudendal nerve can affect erection indirectly through the mechanisms referred.
Also psychiatric mechanisms should be briefly discussed and loss of sensory functions required for sexual desire through pudendal nerve damage. What about the effects on pelvic floor muscles that have accessory function in erectile function? The term neuropraxia should be opened for readers.
3. Radiation therapy options in PCa patients and associated RI-ED rates
The entire sections requires more paragraphs.
The first sentence requires a reference.
IIEF should probably mentioned separately. Prostate Cancer symptom indices usually have only 1-2 questions on sexual function but it may be true that in the lack of better data they are used.
The sentence starting with "Dose escalation..." again lacking a reference. Also "Considering... tumor cells" lacks a reference.
SBRT has been studied in all risk groups not just low and intermediate ( you seem to be familiar with PACE-B but not aware with hypo-rt-pc). Also I would questions referring it as valid due to lack of long-term data. HYPO-RT-PC also had worse QoL scores in SBRT group after the treatment although not in long-term follow-up (due to the worse scores in gastrointestinal PROs). See EAU guidelines for the actual emphasis to which amount it can actually be used (see "Ultra-hypofractionated versus conventionally fractionated radiotherapy for prostate cancer (HYPO-RT-PC): patient-reported quality-of-life outcomes of a randomised, controlled, non-inferiority, phase 3 trial" by Fransson et al.)
I would not include the brachytherapy in this review at all since it is lengthy, and brachytherapy is not in any case discussed in detail. Its ED pathophysiology can differ from EBRT. If this section is included, it is currently lacking references in the first two sentences. Since neither Putora et al. or Bergman et al. were randomized studies the differences could be already present at the baseline and the review does not elucidate this
4. Sexual rehabilitation in RI-ED patients
Again lacking references. I would greatly question the need of a guideline for RI-ED. It can probably be handled within the general erectile dysfunction guidelines, such as EAU's. Although EAU's prostate cancer guideline is referred, EAU's sexual dysfunction guideline is not and would be probably essential when suggesting treatment algorithms. Also, I think that it is unlikely that prostate cancer patients would necessarily benefit from different treatment options, since the proposed pathophysiological mechanisms (nerve damage, atherosclerosis, fibrosis) do not sound very different from those caused by other diseases. At least references and arguments should be given for such a claim.
Again lengthy paragraphs here towards the end.
Pelvic floor physiotherapy should be discussed.
Conclusions:
Nothing else to add except "affect their QoL" should be changed into "their sexual QoL"
Tables and Figures are fine.
What hasn't been said, I think QoL sexual results from ProtecT should be included. As goes for the common knowledge, ED occurs later after RT than in RP which is a significant advantage.
Overall:
The readibility require much editing. Paragraphs are too lengthy. The manuscript would probably benefit omitting sections on brachytherapy altogether. Many sentences are without the proper references.
Overall the scientific value can be questioned after the comprehensive review by Ramirez-Fort et al. released only three years ago. I however understand if the publishes wants to include this theme in the forthcoming theme number. In this case, the review still needs major revisions.
Reviewer 2 Report
Good quality narrative review - it should be mentioned in the title
Several remarks that should be taken into account before accepting for publication:
- The manuscript should be formatted according to the template of Uro for narrative review
- 1. Introduction - no significant remarks
- 2. Pathophysiology - nice presentation on the etiology of RI-ED with all the critical structures involved. Psychological mechanisms of RI-ED must also be mentioned in this paragraph, as well as ADT in the sense of endocrinological pathophysiological mechanism.
- 3. RT alternatives - beneficial is the comprehensive discussion on contemporary EBRT methods and specifically their effects on RI-ED rates - the effect of treatment intensification, hypofractionation, MRI usage in pre-planning, peri-prostatic spacers and intraprostatic markers, SBRT.
There should be a mentioning that probably RI-ED mechanisms differ between BT and EBRT
The discussion on reference 31 should be re-written - hard to understand in the present form - maybe a graph?
- 4. Treatment options on RI-ED - only paragraph on medication (PDE-5) is OK, physiotherapy is only mentioned in the form of LI-ESWT only, as well as regenerative therapies in the form of platelet-rich plasma
- no Future directions Section! - one of the main benefits of narrative reviews - it should be emphasized not only mentioned in the text
As a whole the text is sub-optimally formatted - the paragraphs on different topics are merged together, clumsy and lengthy sentences at several occasions - need a through linguistic improvement
I will recommend a reference to EAU Guidelines on Sexual and Reproductive Health after Figure 1 in the text.
The references are appropriate and relevant to the subject and my recommendation is to accept this manuscript for publication after authors` response on the abovementioned points.
Round 2
Reviewer 1 Report
The authors have revised the manuscript well according to the comments. I found no remaining issues, and therefore I recommend acceptance.